# Solitary-specific drinking to cope motives explain unique variance in solitary drinking behavior but not alcohol problems compared to general drinking to cope motives

**Carillon J. Skrzynski**[1]*, **Kasey G. Creswell**[2]

**1** Department of Psychology and Neuroscience, University of Colorado Boulder, Boulder, CO, United States of America, **2** Department of Psychology, Carnegie Mellon University, Pittsburgh, PA, United States of America

\* cask1436@colorado.edu

## Abstract

**Data Availability Statement:** Data on the relevant variables included in the study manuscript are available from the Open Science Framework database (https://osf.io/9ctnz/).

### Objective

Adolescent and young adult solitary drinking is prospectively associated with alcohol problems, and it is thus important to understand why individuals engage in this risky drinking behavior. There is substantial evidence that individuals drink alone to cope with negative affect, but all prior studies have assessed motives for alcohol use without specifying the context of such use. Here, we directly compared solitary-specific drinking to cope motives with general drinking to cope motives in their ability to predict solitary drinking behavior and alcohol problems. We hypothesized that solitary-specific drinking motives would provide additional predictive utility in each case.

### Methods

Current underage drinkers ($N$ = 307; 90% female; ages 18–20) recruited from a TurkPrime panel March-May 2016 completed online surveys querying solitary alcohol use, general and solitary-specific coping motives, and alcohol problems.

### Results

Both solitary-specific and general coping motives were positively associated with a greater percentage of total drinking time spent alone in separate models, after controlling for solitary-specific and general enhancement motives, respectively. However, the model with solitary-specific motives accounted for greater variance than the general motives model based on adjusted $R^2$ values (0.8 versus 0.3, respectively). Additionally, both general and solitary-specific coping motives were positively associated with alcohol problems, again controlling for enhancement motives, but the model including general motives accounted for greater variance (0.49) than the solitary-specific motives model (0.40).

**Funding:** This work was supported by the National Institutes of Health (grant: R01AA025936 to KGC). The content is solely the responsibility of the authors and does not represent the opinion of NIH. They played no role in study design, data collection and analysis, decision to publish, or preparation of the manuscript.

**Competing interests:** The authors have declared that no competing interests exist.

## Conclusion

These findings provide evidence that solitary-specific coping motives explain unique variance in solitary drinking behavior but not alcohol problems. The methodological and clinical implications of these findings are discussed.

## Introduction

Solitary alcohol consumption is a relatively uncommon drinking pattern among adolescents and young adults, who do the vast majority of their drinking in the company of others (e.g., [1–4]). While most young people do not drink alone, research indicates there is substantial risk associated with solitary drinking for the adolescents (14–27%; [3, 5, 6]) and young adults (~24–40%; [4, 5]) who engage in this drinking behavior [1]. Adolescent and young adult solitary drinking is both cross-sectionally and prospectively associated with greater alcohol consumption and more alcohol problems (e.g., [4, 7–13]). For example, Tucker and colleagues [13] found that 8[th] grade solitary (vs. social-only) drinkers reported greater alcohol use and problems, and 8[th] graders who reported drinking alone went on to endorse more alcohol problems at age 23 after accounting for 8[th] grade alcohol use. Similarly, using large national samples of US adolescents followed for 17 years, Creswell and colleagues [5] recently found that adolescent (modal age 18) and young adult (modal ages 23/24) solitary drinking was concurrently associated with binge drinking and prospectively predicted increased risk for alcohol use disorder (AUD) symptoms at age 35 after controlling for earlier binge drinking, frequency of alcohol use, and a host of other sociodemographic variables (see also [8]). Thus, solitary drinking appears to account for unique variance in predicting later alcohol misuse, above and beyond other risk factors, and may serve as an early warning signal for the development of alcohol problems [1, 4].

In addition to problematic alcohol consumption, solitary drinking is associated with several other negative psychosocial factors in adolescents and young adults, including problems in emotional, social, academic, and legal domains (e.g., [12–15]). For instance, solitary drinkers report more negative affect and social discomfort (e.g., greater social anxiety; lower perceptions of social skills) [4, 16–18], earn poorer grades [13], engage in more violent and deviant acts [15], and experience more problems with authorities than their social-only drinking peers [19]. Because solitary drinking is a risky drinking style, it is critical to know why individuals engage in this behavior to develop effective intervention and prevention programs aimed at helping those most vulnerable to drinking alone.

According to Cox and Klinger's motivational model for alcohol use [20], individuals choose to drink based on expected affective changes that drinking will produce, including increasing positive emotion and/or decreasing negative emotion. In line with the latter, several researchers [3, 4, 8, 10, 21–24], and a recent theoretical model [1], have hypothesized that solitary drinking is motivated by the desire to alleviate negative affect. Solitary drinking is consistently and positively associated with drinking to cope motives (e.g., [6, 10, 12]), even after controlling for social, enhancement, and conformity motives [21, 25, 26]. Additionally, in the laboratory, increases in negative affect in response to a mood manipulation predicted solitary drinking preference relative to social drinking preference among young adults with a history of drinking alone [27]. Solitary drinking is also associated with greater beliefs in alcohol's ability to reduce negative affect [19], lower perceived ability to refuse drinking in the context of negative emotions [19, 28], and drinking in the context of negative affect [8]. Indeed, recent meta-analytic

results indicate a reliable association between adolescent and young adult solitary drinking and a negative reinforcement factor containing measures of motives, expectancies, and contexts related to alleviating negative affect ($r$ = 0.28 [4]). Further, recent cross-sectional [22] and longitudinal [24] studies in young adults found that drinking to cope motives mediated the link between solitary drinking and alcohol problems, even after controlling for enhancement motives. These findings strengthen the hypothesis that drinking to cope is a key mechanism driving the deleterious outcomes of drinking alone [1].

Despite these well-established associations between solitary drinking and drinking to alleviate negative affect, as well as recent studies showing that drinking to cope motives mediate the link between solitary drinking and alcohol problems [22, 24], we are unaware of any prior studies that have assessed motives specifically for drinking alone. That is, to our knowledge, all prior studies have instead assessed motives for alcohol use in general (with no context of alcohol use specified) and then correlated those responses with solitary drinking measures and/or tested whether those responses mediated the link between drinking alone and alcohol problems. It is thus unknown whether coping motives for drinking alone (referred to hereafter as solitary-specific coping motives) better predict solitary drinking behavior and alcohol problems compared to general drinking to cope motives. In short, studies that explicitly ask about solitary drinking motives are necessary to fully understand the reasons for drinking alone [4, 23].

The present study aimed to address this gap in the literature by investigating solitary-specific drinking motives in a sample of 307 underage (i.e., 18–20 year old) drinkers who endorsed drinking alone in the past year. To our knowledge, this is the first study to test assumptions put forth in the literature as to what may specifically drive solitary alcohol consumption, and to determine whether solitary-specific coping motives explain more variance in alcohol problems than general drinking to cope motives. We hypothesized that solitary-specific drinking motives would provide additional predictive utility, compared to general drinking to cope motives, in predicting solitary drinking behavior and alcohol problems, providing evidence that there is added value in assessing coping motives specifically for drinking alone. To be comprehensive, we also assessed solitary-specific enhancement motives, and controlled for these when examining associations between solitary-specific (and general) coping motives and solitary drinking behavior and alcohol problems. Without such analyses, it would be unclear whether the desire to ameliorate negative affect was uniquely associated with solitary drinking and alcohol problems beyond other drinking motives (see also [4, 22, 23, 25, 29]).

## Materials & methods

### Participants

Participants were drawn from a parent study of current underage (i.e., 18–20 year old) drinkers residing in the United States, who were recruited through an Amazon TurkPrime panel from March 2016 to May 2016 (see [12] for additional details). Reliable and valid substance use data has been obtained through such online samples (e.g., [30, 31]). Briefly, among the 727 individuals who were eligible to participate in the study (i.e., 18–20 year old individuals residing in the US who endorsed current drinking [yes/no]), 448 endorsed solitary drinking in the past year. After removing individuals who failed to correctly answer the majority (i.e., $\geq$ 3 out of 4) of attention check questions that were randomly embedded in the survey (e.g., "Select option 3 if you are paying attention"), the final sample size was 307. Most participants ($M_{age}$ = 19.28, $SD_{age}$ = 0.76), were female (89.90%), single (80.46%), and in college (68.40%); 7.17% were in high school. The majority of the sample self-identified as Caucasian (69.38%), while 13.03% identified as African American, 9.77% as multiracial, 4.56% as Asian, 2.28% as American Indian or an Alaska Native, and 0.98% as Native Hawaiian or other Pacific Islander;

84.36% identified as non-Hispanic/Latino. The study and all materials were approved by the Carnegie Mellon University Institutional Review Board. All participants provided written informed consent.

## Measures

**Alcohol consumption.**   Past year alcohol use quantity (standard drinks/occasion) and frequency (days/year) were measured using the National Institute of Alcohol Abuse and Alcoholism's (NIAAA) alcohol consumption question set [32] but were recoded such that higher responses indicated more frequent and heavier alcohol consumption. To assess solitary drinking, participants were asked to indicate the percentage of time that their drinking occurred while alone (i.e., "without anyone else around") versus with others (on a 0–100% scale) in the past year (see [8, 27]) as well as in their lifetime. As noted above, only participants who reported any drinking alone (i.e., >1% of drinking time spent alone) in the past year were included given that results were comparable across both lifetime and past year use, but past year use aligned with the timeframe of our other alcohol consumption and alcohol problem variables.

**General drinking motives.**   Reasons for drinking were assessed with the Drinking Motives Questionnaire-Revised (DMQ-R; [21]), which has demonstrated good criterion validity [33]. Participants were asked how often they drank for each of 20 reasons using a 5-point scale (*almost never/never*, *some of the time*, *half of the time*, *most of the time*, *almost always/always*). Based on study hypotheses, we computed two subscale scores: coping (e.g., "because it helps you when you feel depressed or nervous," $\alpha = 0.87$), and enhancement (e.g., "because you like the feeling," $\alpha = 0.86$).

**Solitary-specific drinking motives.**   Solitary-specific drinking motives were assessed with an adapted version of the Drinking Motives Questionnaire-Revised [21], which included only the coping and enhancement subscales. The instructions were rephrased to indicate that participants should think about how these reasons related to their solitary alcohol consumption. The individual items were edited accordingly (e.g., "You drink **by yourself** to forget your worries"). Reliability was good for both the coping and enhancement subscales ($\alpha = 0.90$ and $0.87$, respectively).

**Alcohol-related problems.**   Alcohol problems in the past year were assessed with the Brief Young Adult Alcohol Consequences Questionnaire (B-YAACQ; [34], which is a 24-item measure that assesses negative consequences associated with alcohol use among young adults. A total score was obtained by summing items, with higher scores reflecting more severe alcohol-related problems [34]. Reliability was good for this measure ($\alpha = 0.91$).

## Data analyses

We first examined descriptive statistics, and all variables had acceptable skewness and kurtosis values (see Supporting Information S1 Table for ranges and skewness/kurtosis values for all study variables). We then used bivariate correlations to assess associations among study variables. Following that, we ran separate linear regression models to directly compare the ability of solitary-specific (vs. general) coping motives in predicting solitary drinking behavior and drinking problems after controlling for demographics and solitary-specific and general enhancement motives, respectively. Models predicting drinking problems also included general and solitary drinking variables to determine the specific effect of coping motives outside of drinking more broadly. We compared models including solitary-specific motives with models including general motives via adjusted $R^2$ values to determine whether solitary-specific coping motive models accounted for more variance in outcomes than models using general

coping motives. This strategy was chosen due to multicollinearity concerns when including both solitary-specific and general drinking to cope motives as predictors in the same model. All analyses were run using R Studio [35].

## Results

Table 1 shows descriptive statistics and bivariate correlations among study variables. As can be seen, variables were positively and significantly associated, with the exception of associations between percentage of past year drinking time spent alone with past year drinking quantity/frequency, general enhancement motives, and B-YAACQ. Associations between general and solitary-specific drinking motives were particularly strong.

Table 2 shows results from linear regression models predicting percentage of drinking time spent alone from solitary-specific (Model 1) and general (Model 2) coping motives. As shown in Model 1, solitary-specific drinking to cope was positively associated with percentage of time spent drinking alone after controlling for demographic variables, while solitary-specific enhancement motives were not. In Model 2, both general drinking to cope motives and enhancement motives were significantly associated with percentage of time spent drinking alone after controlling for demographic variables, though in opposite directions (i.e., greater general coping motives were associated with more solitary drinking, but greater enhancement motives were associated with less solitary drinking). When comparing $R^2$ values across models, Model 1 accounted for a small, but significant proportion of the total variance in percentage of time spent drinking alone, while Model 2 accounted for a smaller proportion of the total variance.

Table 3 shows results from linear regression models predicting alcohol problems from solitary-specific (Model 3) and general (Model 4) coping motives. In Model 3, solitary-specific coping motives were positively associated with alcohol problems after controlling for demographic variables, percentage of time spent drinking alone, solitary-specific enhancement motives, and past year drinking frequency and quantity. This model accounted for significant, moderate variance in drinking problems. However, contrary to prediction, Model 4, which tested general drinking to cope motives, resulted in similar findings and accounted for greater variance in drinking problems.

## Discussion

Solitary drinking is a risky drinking pattern associated with heavier alcohol consumption and more alcohol-related problems, both concurrently and prospectively (e.g., [1, 3, 4, 8, 9, 11, 13]). Self-medication, in which individuals drink alone to alleviate negative affect, is the most compelling theory for solitary drinking [1, 3, 21], and recent studies demonstrate that drinking to cope motives mediate the link between solitary drinking and alcohol problems [22, 24]. However, all prior studies that have demonstrated associations between drinking alone and drinking to cope motives, as well as those showing that drinking to cope motives mediate the link between drinking alone and alcohol problems, have thus far assessed general drinking motives without querying motives specific to drinking alone (see [4, 23]). As such, no prior studies have determined whether solitary-specific coping motives can explain unique variance in solitary drinking behavior and alcohol problems compared to general drinking to cope motives. The current study addressed this theoretically and clinically relevant methodological limitation in a sample of undergraduate drinkers with a history of drinking alone (N = 307).

Results were mixed regarding our hypotheses. As predicted, in participants who reported solitary drinking in the past year, greater endorsement of drinking alone to cope with negative affect was significantly and positively associated with a greater percentage of total drinking

**Table 1. Means (SDs) bivariate correlations of study variables.**

| | Means (SD) | 1. | 2. | 3. | 4. | 5. | 6. | 7. | 8. |
|---|---|---|---|---|---|---|---|---|---|
| 1. Drinking quantity | 3.58 (1.82) | - | | | | | | | |
| 2. Drinking frequency | 5.64 (1.87) | 0.40*** | - | | | | | | |
| 3. Solitary drinking | 32.15 (27.31) | 0.02 | 0.05 | - | | | | | |
| 4. Solitary coping | 2.32 (1.18) | 0.23*** | 0.20*** | 0.29*** | - | | | | |
| 5. Solitary enhance | 2.27 (1.09) | 0.30*** | 0.33*** | 0.20*** | 0.47*** | - | | | |
| 6. General coping | 2.50 (1.08) | 0.29*** | 0.24*** | 0.16** | 0.80*** | 0.39*** | - | | |
| 7. General enhance | 2.87 (1.07) | 0.34*** | 0.35*** | -0.02 | 0.34*** | 0.71*** | 0.49*** | - | |
| 8. YAACQ-B | 6.96 (5.87) | 0.40*** | 0.48*** | 0.05 | 0.45*** | 0.43*** | 0.58*** | 0.51*** | - |

Note:

*$p < 0.05$;

**$p < 0.01$;

***$p < 0.001$

Responses for drinking frequency were labeled such that 3 = once a month, 4 = 2 to 3 times a month, and 5 = once a week; Responses for drinking quantity were labeled such that 3 = 3 to 4 drinks and 4 = 5 to 6 drinks; Solitary drinking = percentage of time spent drinking alone; solitary coping = solitary-specific drinking to cope motives; solitary enhance = solitary-specific drinking to enhance motives; general coping = general drinking to cope motives; general enhance = general drinking to enhance motives; YAACQ-B = Young Adult Alcohol Consequences-Brief.

time spent alone. These associations held after controlling for demographic variables and drinking alone for enhancement motives. Importantly, when compared to a model in which general drinking to cope and enhancement motives were used, the model with solitary-specific motives accounted for more variance. These novel findings show that coping motives specifically for drinking alone can provide additional information about the extent to which individuals are engaging in solitary drinking beyond what can be gleaned from general coping motives for drinking. These results corroborate theory (e.g., [1]), empirical studies (e.g., [24]), and prior reviews and meta-analyses [3, 4, 23] to suggest that drinking to cope is a key mechanism driving solitary drinking behavior.

**Table 2. Summary of linear regression models predicting percentage of drinking time spent alone.**

| | Model 1: Solitary-specific motives $R^2 = 0.08$, $F(6,299) = 5.57$, $p<0.001$ | | | | Model 2: General motives $R^2 = 0.03$, $F(6,300) = 2.57$, $p<0.02$ | | | |
|---|---|---|---|---|---|---|---|---|
| **Variable** | **B** | **S.E.B** | **95% CI** | **β** | **B** | **S.E.B** | **95% CI** | **β** |
| Constant | 39.52 | 39.57 | -38.34, 117.38 | - | 70.52 | 40.13 | -8.45, 149.50 | -t |
| Age | -1.20 | 2.00 | -5.13, 2.74 | -0.03 | -2.07 | 2.03 | -6.07, 1.93 | -0.06 |
| Gender | -1.02 | 5.02 | -10.91, 8.87 | -0.01 | 0.50 | 5.14 | -9.62, 10.62 | 0.01 |
| Race | -5.13 | 3.28 | -11.59, 1.33 | -0.09 | -5.07 | 3.37 | -11.70, 1.56 | -0.09 |
| Parent education | 0.23 | 1.23 | -2.19, 2.64 | 0.01 | 0.10 | 1.26 | -2.39, 2.59 | 0.00 |
| Coping motives | 5.99 | 1.46 | 3.12, 8.86 | 0.26*** | 5.67 | 1.64 | 2.45, 8.89 | 0.23*** |
| Enhancement motives | 2.03 | 1.56 | -1.04, 5.10 | 0.08 | -3.36 | 1.64 | -6.60, -0.13 | -0.13* |

Note:

*$p \le .05$;

**$p \le .01$;

***$p \le .001$

CI = confidence interval; Gender: 0 = female, 1 = male; Race: 0 = non-white, 1 = white; Parent education: 1 = completed grade school or less, 2 = some high school, 3 = completed high school, 4 = some college, 5 = completed college, 6 = graduate or professional school after college.

**Table 3. Summary of linear regression models predicting alcohol problems, controlling for solitary drinking percentage.**

| Variable | Model 3: Solitary-specific motives $R^2 = 0.40$, F(9,296) = 23.42, $p<0.001$ | | | | Model 4: General motives $R^2 = 0.49$, F(9,297) = 33.59, $p<0.001$ | | | |
|---|---|---|---|---|---|---|---|---|
| | **B** | **S.E.B** | **95% CI** | **β** | **B** | **S.E.B** | **95% CI** | **β** |
| Constant | -5.43 | 6.92 | -19.05, 8.20 | - | -6.55 | 6.31 | -18.97, 5.88 | - |
| Age | -0.01 | 0.35 | -0.71, 0.68 | -0.00 | -0.06 | 0.32 | -0.69, 0.58 | -0.01 |
| Gender | -0.17 | 0.89 | -1.92, 1.57 | -0.01 | 0.58 | 0.82 | -1.03, 2.19 | 0.03 |
| Race | 0.53 | 0.57 | -0.60, 1.66 | 0.04 | 0.55 | 0.53 | -0.49, 1.59 | 0.04 |
| Parent education | 0.02 | 0.22 | -0.41, 0.46 | 0.00 | -0.02 | 0.20 | -0.42, 0.38 | -0.00 |
| Past year quantity | 0.52 | 0.17 | 0.20, 0.85 | 0.16** | 0.34 | 0.16 | 0.03, 0.64 | 0.10* |
| Past year frequency | 0.97 | 0.16 | 0.66, 1.29 | 0.31*** | 0.88 | 0.15 | 0.59, 1.17 | 0.28*** |
| Past year solitary | -0.02 | 0.01 | -0.04, 0.00 | -0.09[t] | -0.01 | 0.01 | -0.02, 0.01 | -0.02 |
| Coping motives | 1.54 | 0.26 | 1.02, 2.06 | 0.31*** | 2.16 | 0.26 | 1.65, 2.68 | 0.40*** |
| Enhancement motives | 0.81 | 0.29 | 0.25, 1.38 | 0.15** | 0.98 | 0.27 | 0.44, 1.51 | 0.18*** |

Note:

* $p \le .05$;

** $p \le .01$;

*** $p \le .001$,

[t] $p < 0.10$

CI = confidence interval; Gender: 0 = female, 1 = male; Race: 0 = non-white, 1 = white; Parent education: 1 = completed grade school or less, 2 = some high school, 3 = completed high school, 4 = some college, 5 = completed college, 6 = graduate or professional school after college; Past year quantity = past year average amount of alcohol consumed; Past year frequency = past year average frequency of drinking; Past year solitary = past year percentage of time spent drinking alone.

Replicating prior work [22, 24], we found that drinking alone was not associated with alcohol problems after accounting for solitary-specific drinking to cope motives, which were themselves significantly and positively associated with drinking problems. These findings demonstrate that coping motives at least partially mediate the link between solitary drinking and alcohol problems. Notably, and in contrast to our predictions, we did not find evidence that solitary-specific coping motives explained additional variance in alcohol problems when compared to general coping motives.

These findings have important methodological and clinical implications. Methodologically, these results suggest that measures that assess drinking motives, regardless of the context of such drinking, are sufficient to capture the mechanism driving the association between drinking alone and alcohol problems (i.e., drinking to alleviate negative affect). In contrast, solitary-specific coping motives provided additional information beyond general coping motives about the extent to which an individual engages in solitary alcohol consumption. Clinically, these findings suggest that interventions should target reliance on drinking to regulate negative affect (i.e., coping motives for drinking), regardless of whether such drinking is done alone or with others. However, given reliable associations between drinking to relieve negative affect and solitary drinking [4, 23], it might be useful for clinicians to frame conversations around drinking contexts as a way to explore coping motives for drinking and to identify alternative reinforcement options to target in treatment (e.g., [1, 35–38]). Additionally, because solitary drinking is reliably linked to drinking to cope motives (which are established risk factors for AUD), and since solitary drinking is an easily observable behavior that can be assessed with a simple yes/no question, assessment of solitary drinking may be ideal for brief screening instruments for risky alcohol use in young people with referral to treatment as needed.

It is worth noting that while general enhancement motives were not associated with solitary drinking in bivariate correlations, solitary-specific enhancement motives were. The former is

consistent with findings from prior studies and meta-analyses on adolescents and young adults that have typically not found enhancement motives to be associated with solitary drinking (e.g., [4, 21, 22, 25, 26, 39]), while the latter is unexpected. Notably, though, solitary-specific enhancement motives were not associated with solitary drinking in models accounting for other variables, and general enhancement motives were negatively associated solitary drinking, further supporting that solitary drinking may be more a function of negative reinforcement (i.e., alleviating negative affect) rather than positive reinforcement (i.e., enhancing positive affect). That said, future work is indicated to better understand whether enhancement motives are associated with the extent to which individuals engage in solitary drinking and how much they consume while drinking alone; and, if so, whether these relationships may be driven by particular enhancement motives (e.g., to get drunk) over others (e.g., to enhance positive emotions), the latter of which is thought to be more important in driving social alcohol consumption (see [1]). In this regard, it may be useful to create and validate scales that are specifically focused on motives for solitary drinking and may led to greater understanding of such nuances. Similarly unexpected, bivariate correlations also did not show a significant association between percentage of time spent drinking alone and drinking quantity, frequency or B-YAACQ scores, correlations which have been found in prior work including our own [12]. However, this could be due to a lack variance, as this study was restricted to only those participants who endorsed drinking alone in the past year, while prior work has included social drinkers as well as solitary-only drinkers.

Despite the important implications of this work, it is also important to consider its limitations. One important limitation is that this is a cross-sectional study and thus we cannot speak to the direction of the associations. Additional longitudinal and experimental work is needed to fully understand associations between solitary drinking, coping motives, and alcohol problems (e.g., see [27] for a recent experimental investigation of drinking alone). Further, only the coping and enhancement motive subscales of the DMQ-R were modified to assess solitary-specific reasons for drinking in the current study. Solitary drinking, by definition, precludes social motives for drinking in the moment, but individuals might drink while they are alone before interacting with others to enhance those social experiences or to ameliorate social anxiety (see [18]). Future studies that include solitary-specific social motives are indicated, and again, scales that are tailored to measure such motives for solitary drinking specifically may be useful in providing greater nuance and understanding of this. Finally, since recent work has found that drinking to cope motives mediate the association between depression/suicidal ideation and solitary drinking [40], it would also be interesting to investigate solitary-specific coping motives in such a path model.

An additional limitation is that participants in this study were predominantly white (~70%) and female (~90%). While samples that contain predominantly white and female individuals are not unusual in studies on solitary drinking (e.g., [10, 40, 41]), it is important to note that such results may not generalize to other individuals, and research is clearly needed on more diverse samples. However, a better understanding of solitary drinking among females is particularly needed, given recent findings using nationally representative samples of US high school seniors demonstrating that solitary drinking is increasing among females [6]. Further, studies have shown relatively stronger links between solitary drinking and alcohol problems among females than males [5, 21]. For instance, the odds of AUD symptoms in adulthood were 86% higher for adolescent females who drank alone (vs. social-only drinkers), while the odds were only 8% higher for adolescent males who drank alone (vs. social-only drinkers) [5]. Thus, solitary drinking among females (vs. males) appears to be particularly risky and understanding motives for drinking alone among female drinkers remains an important research goal. A final point of consideration is that this sample reported drinking alone for approximately one third

of their total drinking time; while it is common for young adults to spend more time drinking socially than solitarily [1, 4, 28], findings may have been strengthened if the sample was screened for those who engaged more often in solitary drinking. This is especially important as the variance across models 1 and 2 was small (0.03 and 0.08) indicating that, at least among this sample, the included variables only explained a small proportion of solitary drinking. Future studies that focus on individuals who primarily drink by themselves may shed further light on solitary drinking motives and their relation to solitary alcohol consumption and alcohol problems.

Despite these limitations, this study makes an important contribution to the literature on solitary drinking in that it reflects more precisely the motivation individuals have for engaging in solitary alcohol consumption. Our findings showing that solitary-specific coping motives account for unique variance in solitary drinking behavior, relative to general coping motives, support theoretical accounts of solitary drinking that emphasize the desire to alleviate negative affect as the primary reason for drinking alone. The fact that solitary-specific coping motives did not provide additional explanatory power for predicting alcohol problems over general drinking to cope motives suggests that the usefulness of solitary-specific drinking motives varies depending on the outcome. In general, the antecedents and consequences of solitary drinking remain important topics of research and require further study.

## Supporting information

**S1 Table. Ranges and skewness/kurtosis values for all study variables.**
(PDF)

## Acknowledgments

The authors would like to thank the students and staff of the Behavioral Health Research Lab at Carnegie Mellon University as well as the participants who volunteered their time to this research.

## Author Contributions

**Formal analysis:** Carillon J. Skrzynski.

**Funding acquisition:** Kasey G. Creswell.

**Resources:** Kasey G. Creswell.

**Supervision:** Kasey G. Creswell.

**Writing – original draft:** Carillon J. Skrzynski.

**Writing – review & editing:** Carillon J. Skrzynski, Kasey G. Creswell.

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
