## [Decision Letter · Decision Letter 0]

20 Jan 2023

PONE-D-22-20380Solitary-specific drinking to cope motives explain unique variance in solitary drinking behavior but not alcohol problems compared to general drinking to cope motives

Dear Dr. SKRZYNSKI

Thank you for submitting your manuscript to PLOS ONE. After careful consideration, we feel that it has merit but does not fully meet PLOS ONE’s publication criteria as it currently stands. Therefore, we invite you to submit a revised version of the manuscript that addresses the points raised during the review process.

ACADEMIC EDITOR: Minor revision required.

We look forward to receiving your revised manuscript.

Kind regards,

Anthony A. Olashore, MBCHB, FWACP

Academic Editor

PLOS ONE

Journal Requirements:

“This work was supported by the National Institutes of Health (grant: R01AA025936 to KGC). The content is solely the responsibility of the authors and does not represent the opinion of NIH. They played no role in study design, data collection and analysis, decision to publish, or preparation of the manuscript.”

Please include your amended Funding Statement within your cover letter. We will change the online submission form on your behalf**.**

Reviewers' comments:

Reviewer's Responses to Questions

1. Is the manuscript technically sound, and do the data support the conclusions?

Reviewer #1: Yes

Reviewer #2: Yes

2. Has the statistical analysis been performed appropriately and rigorously? 

Reviewer #1: Yes

Reviewer #2: Yes

3. Have the authors made all data underlying the findings in their manuscript fully available?

Reviewer #1: Yes

Reviewer #2: Yes

4. Is the manuscript presented in an intelligible fashion and written in standard English?

Reviewer #1: Yes

Reviewer #2: Yes

5. Review Comments to the Author

Reviewer #1: This manuscript describes the results of a study examining solitary-specific drinking to cope motives, as compared to general drinking to cope motives using data obtained from an online survey sample of underage drinkers. The authors examined both alcohol use behaviors and alcohol problems as outcomes. Results indicated that both solitary-specific coping motives and general coping motives were positively associated with the percent of total drinking time spent alone and with alcohol problems in separate models. Solitary-specific coping motives models accounted for greater variance in alcohol use than the general motives model. Conversely, general coping motives accounted for greater variance in alcohol problems than the solitary-specific coping motives. The authors include a thoughtful discussion of the methodological and clinical implications of these findings.

This is an interesting study that follows previous work by the authors examining solitary drinking behavior, which has been shown by them, and others, to be a high-risk behavior. The study has a nuanced hypothesis that helps to fill the critical gap in identifying the motives – solitary-specific vs. general – for using alcohol to cope. The study is well-described with detailed methods and results. The manuscript is well-written and easy to follow. The discussion is logical and thoughtful and balances the strengths and limitations and future directions of the work. There are a couple of issues of note:

1) Given the nuanced nature of the hypothesis and objectives, it was interesting to see that authors tried to separate the motives – solitary-specific vs. general – in predicting the proportion of time spent in solitary drinking. The range of solitary drinking time reported by the individuals seems broad but on the low end. I was wondering about how the results might be different (strengthened?) if the authors had screened for a sample with greater prevalence of solitary drinking. Perhaps the authors can mention this potential limitation and consider this as a future direction.

2) Another way to examine the relationships being pursued in this manuscript would be to contrast the motives for drinking alone vs. drinking with others. This was particularly suggested by the finding that solitary-specific enhancement motives were a significant predictor of solitary drinking. Perhaps the authors considered modeling the proportion of time spent drinking with others as a function of the drinking motives – general vs. solitary-specific – to get additional insights on the predictors of the two different drinking patterns. Given that the sample had a mixed drinking pattern (alone as well as with others), there might be some interesting insights to be gleaned by comparing the results of complementary analyses of solitary drinking time and of drinking time with others. Perhaps this is beyond the scope of this manuscript and/or a future direction of the authors.

Thank you for the opportunity to review this interesting work!

Reviewer #2: Dear Authors,

Thank you for submitting this interesting manuscript which extends our understanding of the interface of solitary drinking (and specific motives), drinking motives and actual alcohol use and problems. It is well situated in the growing literature and has implications for clinical practice.

While largely a very strong manuscript for recommendation to be published, the following may be incorporated:

Intro

1. Augment the recent theoretical model (Creswell, 2021) with the seminal motivational model of alcohol (Cox & Klinger, 1988) for greater grounding.

Methods

2. Insert the time period of data collection

3. I am unsure of the value of further describing participants as "84.36% identified as non-Hispanic/Latino". Perhaps there is a social significance unknown to me and others who would benefit for some commentary related to this description.

Results

4. At line 209, what is labelled as Model 2 should be Model 4.

Discussion

5. While Models 1 & 2 are significant, the variance (8% and 3%) accounted by the models requires tentative discussion to avoid overplaying the effect of the included variables.

Abstract

6. Include gender distribution of participants and time/dates of data collection to aid readers appraise the study from the get go.

8. Include the magnitude of the variance explained by models.

---

## [Author Response · Author response to Decision Letter 0]

31 Jan 2023

Reviewer #1: 

1) Given the nuanced nature of the hypothesis and objectives, it was interesting to see that authors tried to separate the motives – solitary-specific vs. general – in predicting the proportion of time spent in solitary drinking. The range of solitary drinking time reported by the individuals seems broad but on the low end. I was wondering about how the results might be different (strengthened?) if the authors had screened for a sample with greater prevalence of solitary drinking. Perhaps the authors can mention this potential limitation and consider this as a future direction.

This is a good point, and we have now included it in the discussion section on pages 15 and 16: “A final point of consideration is that this sample reported drinking alone for approximately one third of their total drinking time; while it is common for young adults to spend more time drinking socially than solitarily [1,4,42], findings may have been strengthened if the sample was screened for those who engaged more often in solitary drinking. This is especially important as the variance across models 1 and 2 was small (0.03 and 0.08) indicating that, at least among this sample, the included variables only explained a small proportion of solitary drinking. Future studies that focus on individuals who primarily drink by themselves may shed further light on solitary drinking motives and their relation to solitary alcohol consumption and alcohol problems.”

2) Another way to examine the relationships being pursued in this manuscript would be to contrast the motives for drinking alone vs. drinking with others. This was particularly suggested by the finding that solitary-specific enhancement motives were a significant predictor of solitary drinking. Perhaps the authors considered modeling the proportion of time spent drinking with others as a function of the drinking motives – general vs. solitary-specific – to get additional insights on the predictors of the two different drinking patterns. Given that the sample had a mixed drinking pattern (alone as well as with others), there might be some interesting insights to be gleaned by comparing the results of complementary analyses of solitary drinking time and of drinking time with others. Perhaps this is beyond the scope of this manuscript and/or a future direction of the authors.

We agree with the Reviewer and in fact our measure of solitary drinking was a continuous scale that assessed the proportion of time spent drinking alone versus with others. As such, this one item includes information on both solitary and social drinking (e.g., if someone said they spent 15% of their time drinking alone, 75% of their time was necessarily spent drinking with others). Thus, the analyses already are a function of both solitary and social drinking and cannot be run separately. 

Thank you for the opportunity to review this interesting work!

We appreciate this nice comment!

Reviewer #2: 

Intro

1. Augment the recent theoretical model (Creswell, 2021) with the seminal motivational model of alcohol (Cox & Klinger, 1988) for greater grounding.

We now briefly describe the Cox and Klinger motivational model to provide grounding for the theoretical account made by Creswell and others on page 4: “According to Cox and Klinger’s motivational model for alcohol use [20], individuals choose to drink based on expected affective changes that drinking will produce including increasing positive emotion and/or decreasing negative emotion. In line with the latter, several researchers [3,4,8,10,21–24], and a recent theoretical model [1], have hypothesized that solitary drinking is motivated by the desire to alleviate negative affect.”

Methods

2. Insert the time period of data collection.

We now note on page 6 the time period over which data was collected: “Participants were drawn from a parent study of current underage (i.e., 18-20 year old) drinkers residing in the United States, who were recruited through an Amazon TurkPrime panel from March 2016 to May 2016 (see [12] for additional details).”

3. I am unsure of the value of further describing participants as "84.36% identified as non-Hispanic/Latino". Perhaps there is a social significance unknown to me and others who would benefit for some commentary related to this description.

We report percentage of non-Hispanic/Latino individuals as this is a separate construct from race and further contextualizes the sample. Additionally, including information on Hispanic/Latino percentages is standard in the literature (e.g., Buckner & Terlecki, 2016; Gonzalez & Halvorsen, 2021; Terry-McElrath, O’Malley, Pang, & Patrick, 2022; Wardell, Kempe, Rapinda, Single, Bilevicius, Frohlich, Hendershot, & Keough, 2020). 

Results

4. At line 209, what is labelled as Model 2 should be Model 4.

We appreciate the Reviewer’s attention to detail. We have corrected this label to read Model 4.

Discussion

5. While Models 1 & 2 are significant, the variance (8% and 3%) accounted by the models requires tentative discussion to avoid overplaying the effect of the included variables.

We now provide a brief discussion of the variance of models 1 and 2 on pages 15-16, noting that the included variables only explained a small proportion of solitary drinking (see also our response above to Reviewer 1, comment 1).

Abstract

6. Include gender distribution of participants and time/dates of data collection to aid readers appraise the study from the get go.

We have added gender distribution and time of data collection to the abstract, as suggested.

8. Include the magnitude of the variance explained by models.

We have added this to the abstract, as well.

---

## [Editor Report · Decision Letter 1]

17 Feb 2023

Solitary-specific drinking to cope motives explain unique variance in solitary drinking behavior but not alcohol problems compared to general drinking to cope motives

PONE-D-22-20380R1

Dear Dr. SKRZYNSKI

We’re pleased to inform you that your manuscript has been judged scientifically suitable for publication and will be formally accepted for publication once it meets all outstanding technical requirements.

Kind regards,

Anthony A. Olashore, PhD, FWACP

Academic Editor

PLOS ONE

---

## [Editor Report · Acceptance letter]

23 Feb 2023

PONE-D-22-20380R1 

Solitary-specific drinking to cope motives explain unique variance in solitary drinking behavior but not alcohol problems compared to general drinking to cope motives 

Dear Dr. Skrzynski:

I'm pleased to inform you that your manuscript has been deemed suitable for publication in PLOS ONE. Congratulations! Your manuscript is now with our production department. 

Kind regards, 

on behalf of

Dr. Anthony A. Olashore 

Academic Editor

PLOS ONE